# Laser Welding of ASTM A553-1 (9% Nickel Steel) (PART I: Penetration Shape by Bead on Plate)

**Jaewoong Kim [1], Jisun Kim [1,\*], Sungwook Kang [2]**  **and Kwangsan Chun [3]**

[1]  Smart Manufacturing Process R&D Group, Korea Institute of Industrial Technology, Gwangju 61012, Korea; kjw0607@kitech.re.kr

[2]  Transport Machine Components R&D Group, Korea Institute of Industrial Technology, Jinju 52845, Korea; swkang@kitech.re.kr

[3]  Welding Engineering R&D Department, Industrial Application R&D Institute, Daewoo Shipbuilding & Marine Engineering Co., LTD., Geoje, Gyeongnam 53302, Korea; kschun@dsme.co.kr

\*  Correspondence: kimjisun@kitech.re.kr; Tel.: +82-62-600-6302

**Abstract:** The International Maritime Organization (IMO) is tightening regulations, in order to reduce greenhouse gas emissions from ship operations. As a result, the number of vessels using Liquefied Natural Gas (LNG) as fuel has increased rapidly. At this time, ASTM A553-1 (9% nickel steel) is being used as a tank material for storing LNG as fuel, because it has higher strength than other cryogenic materials. Currently, shipyards are manufacturing LNG fuel tanks by using the Flux Cored Arc Welding (FCAW) method, using 9% nickel steel material. However, fabrication through FCAW welding has two drawbacks. The first is to use a welding electrode that is 20 times higher in cost than the base metal, and the second is that the total production cost increases because the thickness of the tank increases due to the strength drop near the Heat Affected Zone (HAZ) after welding. Laser welding, which does not require additional welding rods and has no strength reduction in the HAZ, can overcome the drawbacks of FCAW welding and ensure price competitiveness. In this study, it is confirmed the characteristics of the penetration shape of Bead on Plate (BOP) after various laser welding conditions as a basic study to apply laser welding to A553-1 welding. For this, penetration characteristics of A553-1, according to laser welding speed and power, which is a main factor of laser welding, are confirmed.

**Keywords:** laser welding; ASTM A553-1 (9% nickel steel); penetration shape; Bead on Plate (BOP)

## 1. Introduction

With the global environmental regulations to prevent climate change, regulations on emissions of ships are intensifying. In the shipbuilding industry, the use of Liquefied Natural Gas (LNG) for marine fuel is continuously studied as an alternative to existing fossil fuels [1–5]. By using LNG as fuel, $CO_2$ emission is reduced by about 20%, nitrogen oxides ($NO_x$) by 80%, sulfur oxides ($SO_x$) by 90% and particulate matter (PM) by 99%, in comparison to using Heavy Fuel Oil (HFO), which is the existing marine fuel. It enables Tier III compliance, which is the latest regulation from International Maritime Organization (IMO). In other words, in the case of ships using LNG as fuel, it is possible to operate eco-friendly vessels without the need for additional equipment, to reduce harmful exhaust gas when operating.

A total of four materials (9% Nickel, STS 304L, Al 5083-0 and Invar) are available for the low-temperature/cryogenic energy storage/transport vessels approved by International Code of the Construction and Equipment of Ships Carrying Liquefied Gases in Bulk (IGC Code) [6]. Moreover, mechanical properties are shown in Table 1.

**Table 1.** IGC Code list of cryogenic fuel tank material.

| Material | Chemical Composition | Yield Strength (MPa) | Ultimate Tensile Strength (MPa) |
|---|---|---|---|
| 9% Nickel | Fe-9Ni | >585 | 690–825 |
| STS304L | Fe-18.5Cr-9.25Ni | >205 | >585 |
| Al5083-0 | Al-4.5Mg | 124–200 | 276–352 |
| Invar | Fe-36Ni | 230–350 | 400–500 |

A553-1 is used as a material for cryogenic tanks, such as LNG, and has recently been used as an LNG fuel tank material because of its relatively higher yield strength/tensile strength than other materials. There are several factors that determine the thickness of LNG fuel tanks, among which the minimum yield/tensile strength is an important factor. A553-1 has a weak point in that the thickness of the tank becomes thicker due to weakened strength at the weld after welding [7]. When laser welding is applied, the thickness of the tank can be reduced by increasing the strength of this part. The base material used in this study is A553-1, and the chemical composition is shown in Table 2.

**Table 2.** The chemical composition of ASTM A553 Type 1 (9% nickel steel).

| Component | Percentage (wt.%) |
|---|---|
| Carbon, C | 0.13 max |
| Manganese, Mn | 0.90 max |
| Phosphorous, P | 0.015 max |
| Silicon, Si | 0.15–0.40 |
| Sulfur, S | 0.015 max |
| Nickel, Ni | 8.5–9.5 |

Laser is the abbreviation of "Light Amplification by Stimulated Emission of Radiation" and refers to the welding method applying the output of the laser beam. Laser welding has been studied for application to various materials due to its low welding deformation, easy automation and its deep, narrow heat-affected part [8–22]. The purpose of this study was to analyze the penetration characteristics of A553-1, according to the welding speed and power among the laser-welding parameters, and to improve the price competitiveness, by securing the weldability with good mechanical strength in the future.

## 2. Experiments of ASTM 553-1 (9% Nickel Steel) Bead on Plate (BOP) by Laser Welding

### 2.1. Laser-Welding Equipment, Parameters of Experiment and Base Material

For the experiment, 5 kW fiber laser welder were used, and Figure 1 shows the laser-welding oscillator, incidental material, optical system and jig used in the experiment.

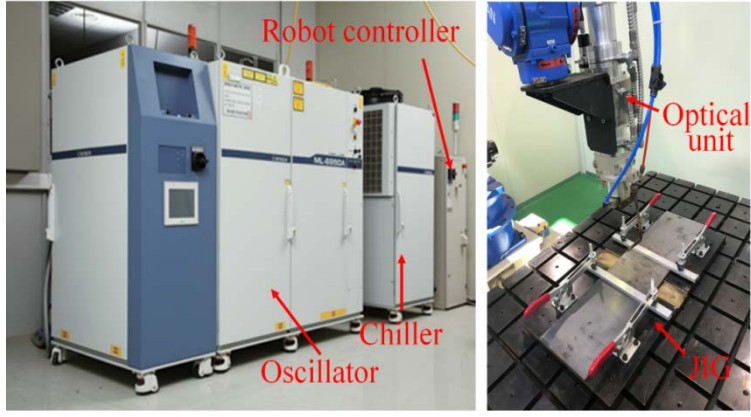

**Figure 1.** Fiber laser welding equipment.

The optical system used in this experiment has a spot diameter of 400 μm. The focus focal length was 148.8 mm, and the depth of focus was 6 mm.

Functionally controllable parameters in laser welding are approximately four parameters (welding speed, laser power, focus position and polarization), and most are optimized for the four conditions and used according to the situation. In addition, the alignment condition (thickness, gap and mismatch) of the welding material is also one of influences on the welding quality. The effect of each welding variable on welding quality is as follows:

(1) Laser power: The laser power is the most important factor in defining the penetration depth limit, and it is impossible to obtain a weld with a certain depth or more, even if the other conditions are optimal.

(2) Welding speed: In addition to the laser output, the welding speed has a close relationship with the welding heat input received by the welding position.

(3) Focus position (defocus): The focus position in the thickness direction is the point of impurity which places the energy focusing point at a specific position in the thickness direction of the material to be welded.

(4) Polarization (work angle and tilling angle): Optimization should be performed by adjusting the working angle and tilting angle according to the shape of the welding seam. In addition, welding must be performed at a constant angle, to protect the equipment according to the reflected light or absorption rate of the material.

(5) Alignment (thickness, gap and mismatch): Depending on the thickness of the welding material, the range of selection of the welding power is determined, since the welding power is related to the penetration depth. In laser welding, gaps play an important role in weld quality (bead shape, so it is important to select welding conditions according to gaps.

In this study, laser power and laser welding speed, which greatly affect the penetration, are controlled, and the penetration shape is observed after BOP.

Experiments are conducted to analyze the effect of welding conditions on the formation of the molten part and the shape of the bead through the BOP test. The penetration depth that can be seen through this experiment provides data that can correspond to the thickness of the base metal. The data on the width of the bead indicate the welding conditions that can be applied to open-gap welds. In order to achieve the above goal, experiments are performed with 2 parameters (laser power and welding speed) and collect data of bead geometry, penetration, micro image and effect of parameters. Each experimental BOP condition for A553-1 using the laser welding process is shown in Table 3.

**Table 3.** Experimental conditions of laser Bead on Plate (BOP).

| Welding Parameters | Experimental Conditions |
|---|---|
| Laser power (kW) | 3.0–5.0 (5 cases) |
| Welding speed (meter per minute, m/min) | 0.3–3.0 (11 cases) |
| Defocus (mm) | 0 |
| Shielding gas | $N_2$, 15 L/min |
| Tilting angle, Working angle | 0° |

In order to analyze the laser-welding characteristics of A553-1, reference data for general laser welding are obtained through the welding experiments of the A36 (low carbon steel) which chemical composition is shown in Table 4 with some of same welding conditions. In this study, the laser weldability of carbon steel is not discussed in detail, but is used only as data to compare welding characteristics of simple A553-1.

**Table 4.** The chemical composition of A36 (low carbon steel).

| Component | Percentage (wt.%) |
| --- | --- |
| Carbon, C | 0.29 |
| Iron, Fe | 98.0 |
| Manganese, Mn | 0.80–1.20 |
| Phosphorous, P | 0.045 |
| Silicon, Si | 0.15–0.40 |
| Sulfur, S | 0.050 |
| Copper, Cu | >0.20 |

### 2.2. Measurement of Bead Geometry

Macro cross-sectional inspection is a method to smoothly polish the surface of a welded part and perform a chemical solution treatment, to examine the structure, pore, penetration, HAZ and the like. Bead geometry measurements are made in all experiments and basically performed in BOP test, butt and fillet welding.

In this study, in order to analyze the effect of welding variables on the shape of the molten part shape, 9 measuring positions were selected, as shown in Figure 2.

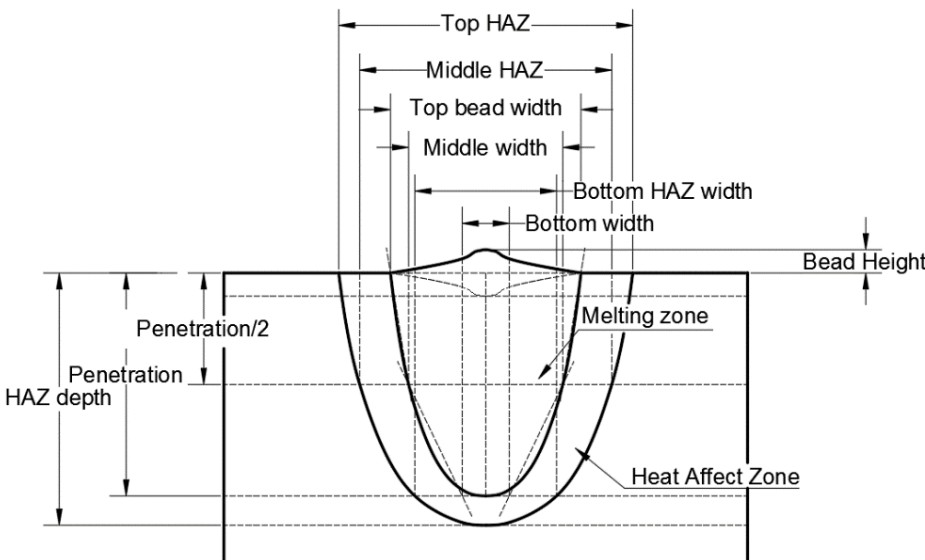

**Figure 2.** Definition of measurement site on bead cross-section.

A description of the cross-sectional measurement positions of the melted zone and the heat-affected zone is as follows:

(1) Top bead width: melted surface width of base material that can be observed with naked eyes.
(2) Top HAZ: the length of the HAZ of the base material surface (Top) observed through the micro-section.
(3) Penetration: the vertical depth of melting zone from surface of base metal.
(4) HAZ Depth: the vertical depth of HAZ from surface of base metal.
(5) Bead Height: the vertical length of the portion protruding above the surface of the base material after the melted portion is formed.
(6) Middle Width: the width of the melting zone at the midpoint of the penetration depth.
(7) Middle HAZ: the width of the HAZ at the midpoint of the penetration depth.
(8) Bottom Width: the width of the melting zone at the endpoint of the penetration depth.
(9) Bottom HAZ: the width of the HAZ at the endpoint of the penetration depth.

In this study, the size of welding specimen is 150 mm × 300 mm × 15mm. The laser power is stabilized at the center of the specimen; the sections from this location make it possible to show the representative section, as shown Figure 3. For measuring back-bead geometry, middle parts of horizontal axis of the experiment specimen were cut in 10 mm × 25 mm size, by a wire-cutting machine, and polished.

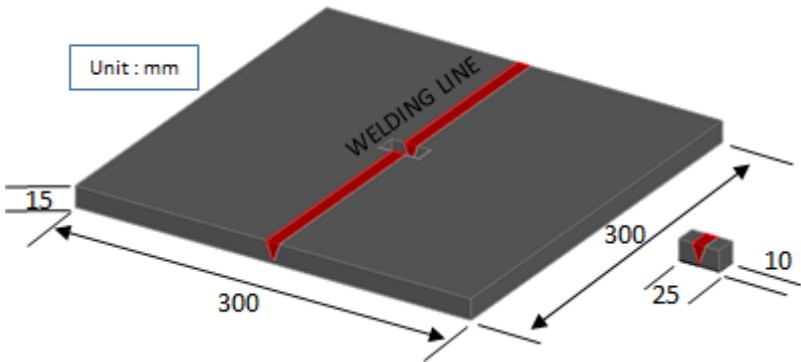

**Figure 3.** Welding specimen and coupon for cross-section observation.

To make the experiment specimen's bead geometry clearly visible, Nital (10% HNO$_3$ and Ethanol) solution were applied for the etching of the cross-section of specimens. Moreover, an optical microscope system was used for accurate measurement of bead geometry and actually measured cross-sectional bead geometries. Figure 4 shows the digital electronic microscope with 2 Mega pixels and photographed at ×60 magnification, associated with bead geometry measurement. The shape of the bead cross-section, the penetration depth, bead width and height are measured by matching the 0.1 mm mesh of the optical camera with the image.

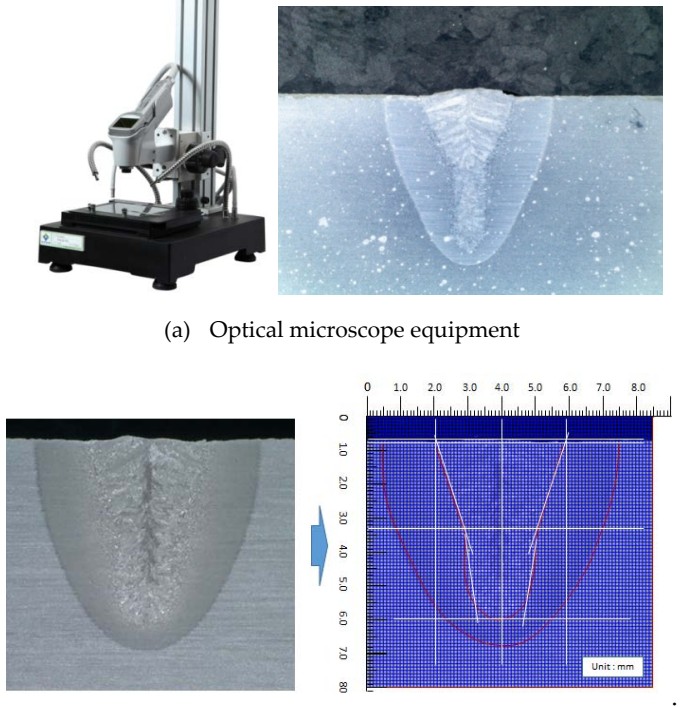

(a) Optical microscope equipment

(b) Bead geometry measurement

**Figure 4.** Equipment and method for measuring for bead geometry; (**a**) Optical microscope equipment, (**b**) Bead geometry measurement.

## 3. Results and Discussions

### 3.1. Measurement of Bead Penetration Shape

In this study, laser-welding power was experimented with on five cases (3, 3.5, 4, 4.5 and 5 kW), and laser-welding speed was experimented with 11 cases (0.3, 0.5, 0.8, 1.0, 1.2, 1.5, 1.8, 2.0, 2.2, 2.5 and 3.0 m/min) for A553-1. The total number of experiments was 55.

Figure 5 shows the results when the laser power was 3 kW and the welding speed was 0.5, 1.0, 1.5, 2.0 and 3.0 m/min, respectively.

| Laser Power: 3 kW | | | | |
|---|---|---|---|---|
| Speed: 3 m/min | Speed: 2 m/min | Speed: 1.5 m/min | Speed: 1 m/min | Speed: 0.5 m/min |

**Figure 5.** Bead geometry on BOP welding test (laser power: 3 kW).

Figure 6 shows the results when the laser power was 4 kW and the welding speed was 0.5, 1.0, 1.5, 2.0 and 3.0 m/min, respectively.

| Laser Power: 4 kW | | | | |
|---|---|---|---|---|
| Speed: 3 m/min | Speed: 2 m/min | Speed: 1.5 m/min | Speed: 1 m/min | Speed: 0.5 m/min |

**Figure 6.** Bead geometry on BOP welding test (laser power: 4 kW).

Figure 7 shows the results when the laser power was 4 kW and the welding speed was 0.5, 1.0, 1.5, 2.0 and 3.0 m/min, respectively.

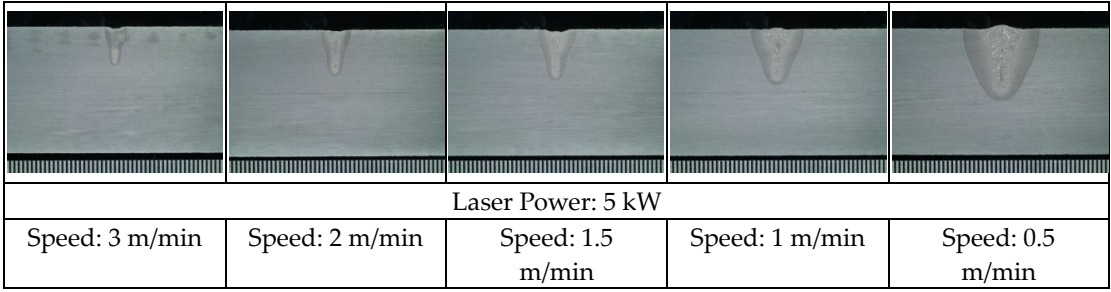

| Laser Power: 5 kW | | | | |
|---|---|---|---|---|
| Speed: 3 m/min | Speed: 2 m/min | Speed: 1.5 m/min | Speed: 1 m/min | Speed: 0.5 m/min |

**Figure 7.** Bead geometry on BOP welding test (laser power: 5 kW).

In the case of the A553-1, which was the focus of this study, micro-cracks were observed in the center line of the welded section with the welding conditions of 5 kW power and 0.5 m/min speed. The internal porosity was confirmed, as shown on the left side of Figure 8. It is a pore that often occurs because of excessive heat, incomplete penetration, gravity-laser angle, etc. [23]. On the contrary,

as shown on the right side of Figure 8, unsafe infusion occurred with the welding speed of 2.5 m/min, and the fine particles melted in an unstable molten state spatter.

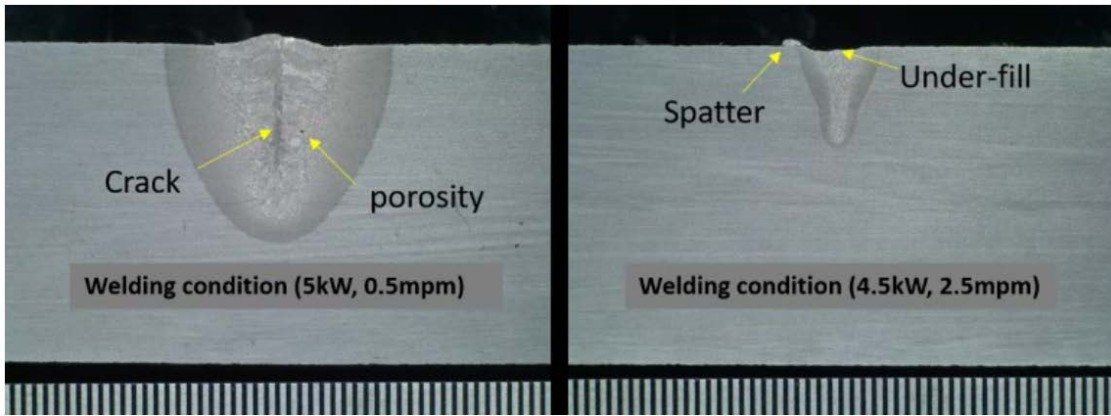

**Figure 8.** Crack, porosity, spatter and under-fill of A553-1 after laser welding.

### 3.2. Effect of Laser Power

In laser welding, laser power is one of the important control variables. It has been reported that the shape of the fused portion is controlled according to the laser power. In this study, its influence on three bead shape was analyzed. The difference in the shape of the beads, according to the laser power, is shown from Figures 9–11.

In the case of bead height, shown in Figure 9, the increase in bead height of A553-1 was not uniform as the laser power increased. Generally, as the welding power increases, the surface beads are under-filled, due to an increase in the amount of spatter generated. However, this result does not necessarily indicate under-fill as the laser power increases. Moreover, when compared with A36, it is confirmed that the height of bead is similar to that of laser power, despite the difference of physical properties of the two metals. There is variation in some sections (3.5 kW), but it can be concluded that the overall trend is almost consistent. It can be predicted that the laser power of both metals has no influence on the height of the weld bead.

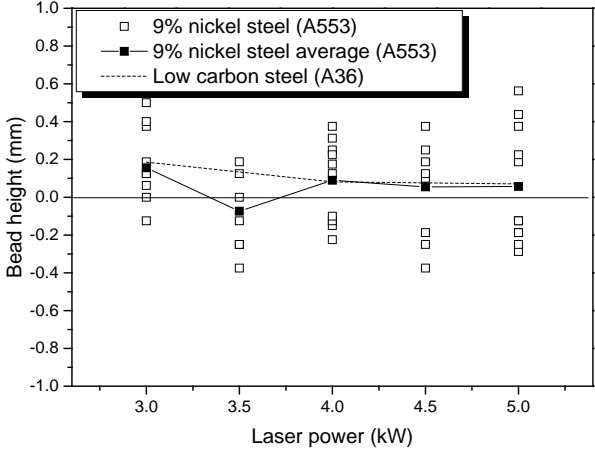

**Figure 9.** Height fluctuation of surface beads, according to laser power.

For A553-1 (the width of the surface bead shown in Figure 10), the width of the bead tends to widen as the laser power increases in the range of 3.5 to 4.5 kW. However, it does not continuously increase in proportion to the laser. In comparison with A36, on average, the bead width of A36 is narrower than the bead width of A553-1. This can be inferred from the fact that the direction of diffusion

of the welding heat source is rapidly diffused in the thickness direction of the base material, so that the formation of the welded keyhole is very fast. The difference in the thermal conductivity coefficient of the two materials can be evidence of the above reasoning. The difference in thermal conductivity coefficient between the two materials is more than twice that of A36. Thermal conductivity coefficient of A553-1 and A36 are 17.43 and 52.7 (W/(m$^2$K)), respectively. When a heat source is supplied to material of A36 with a relatively high thermal conductivity characteristic, heat is transferred in all directions of the base material. It accelerates the continuous penetration of the heat source in the thickness direction of the base material. However, 9% nickel steel (A553) has low thermal conductivity, and as the energy source is concentrated on the surface, the width of the surface beads increases. In conclusion, it is evident that laser power is not an important parameter for the width of the bead.

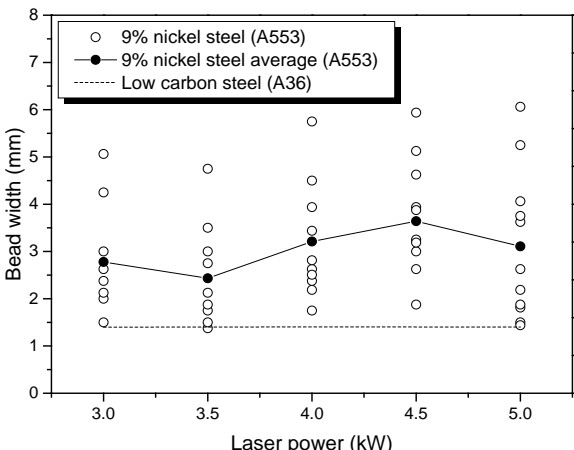

**Figure 10.** Width fluctuation of surface beads, according to laser power.

The most important penetration depth shown in Figure 11 is directly related to welding efficiency. If it is possible to obtain deep penetration at low power, it is very advantageous in terms of productivity. It is also important to determine what conditions need to be adjusted to ensure sufficient penetration depth. As shown in Figure 11, the relationship between penetration depth and laser power shows that the penetration depth linearly increases with increasing laser power. By comparing the penetration-depth changes of the two materials, it can be confirmed that two materials penetrations are formed at almost the same level. The penetration depth of the two materials is highly dependent on the laser power, despite the difference in properties of the two materials.

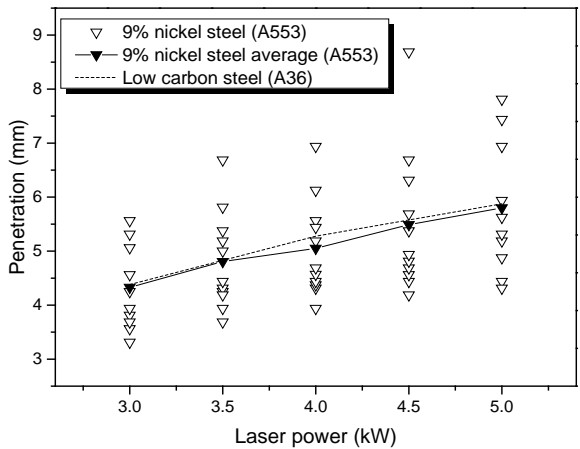

**Figure 11.** Penetration fluctuation of surface beads, according to laser power.

Figure 12 shows that penetration depth increases of A553-1 with increasing output at the same welding speed (1.8 m/min) and same defocus (0, surface). This result shows that laser-power control is required to control the penetration depth efficiently. Further, since the influence of the laser power on the width and height of the surface beads is small, it can be a means for independently controlling the penetration depth.

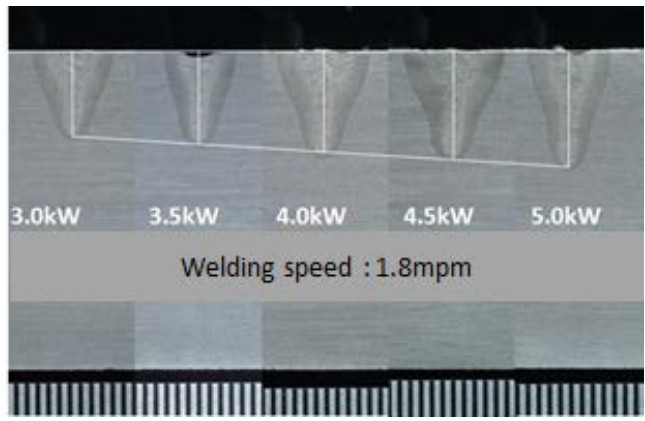

**Figure 12.** Variation of penetration depth, according to laser-power change by macro cross-sectional shape analysis.

### 3.3. Effect of Laser Speed

It is found that the laser power at A553-1 and A36 welds is related only to the penetration depth and does not affect the melt width and height.

However, as can be seen from Figures 13–15, as the welding speed increases, the bead height, width and penetration depth are reduced. In particular, the change in welding speed shows only the characteristics of A553-1.

Figure 13 clearly shows the difference in bead height between A36 and A553-1. As the welding speed increases, A553-1 continues to decrease in bead height, resulting in under-fill. In the case of A553-1, surface beads without under-fill are formed at a speed of 1.2 m/min or higher. On the other hand, at the high speed of 1.5 to 3.0 m/min, it is confirmed that the under-fill appeared as an average number of times. However, in the case of A36, under-fill does not occur, but the bead of uniform height is formed even when the welding speed is changed. A36 also under-fill under certain conditions, but on average, it proves the above. The occurrence of under-fill is described later. In this way, A553-1 can confirm the formation of bead height, depending on the welding speed.

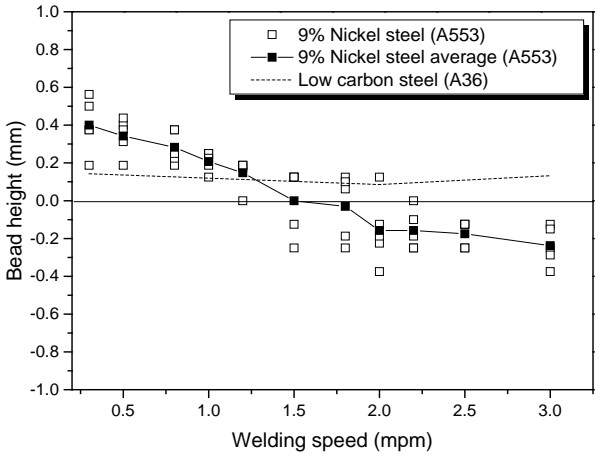

**Figure 13.** Height fluctuation of surface beads, according to welding speed.

In the case of the width of the weld bead shown in Figure 14, A553-1 showed a tendency to decrease as the height of the bead changes, with the change of the welding speed, but it is confirmed that the A36 is kept constant despite the change of the welding speed. This is the laser-welding characteristic ofA553-1, and it is a result of proving that the welding speed is one of the important parameters for bead width control of A553-1.

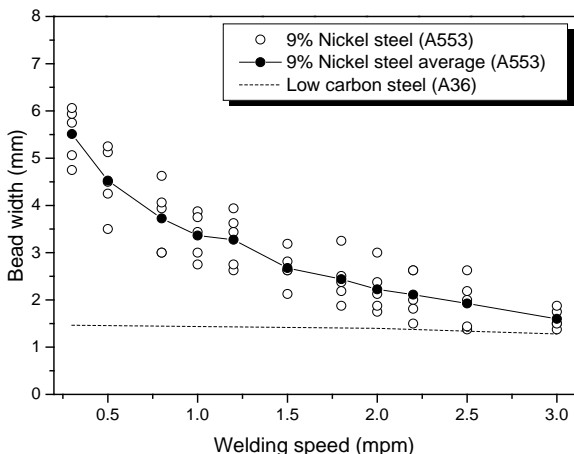

**Figure 14.** Width fluctuation of surface beads, according to welding speed.

For the depth of penetration shown in Figure 15, both A553-1and A36 have the same results. As the welding speed increases, it is similar that the penetration depth decreases in both materials, but on average, it is confirmed that the penetration depth of A36 is formed deeper under the same welding conditions. These results show that the welding speed is controlled to control the penetration depth in general A36 welding. However, in the case of A553-1, it is possible to control both the welding depth and the width and height of the surface bead. This is evidence that welding speed has a large impact on heat input in A553-1 laser welding. This is because the size of the melted portion decreases proportionally as the amount of heat input decreases.

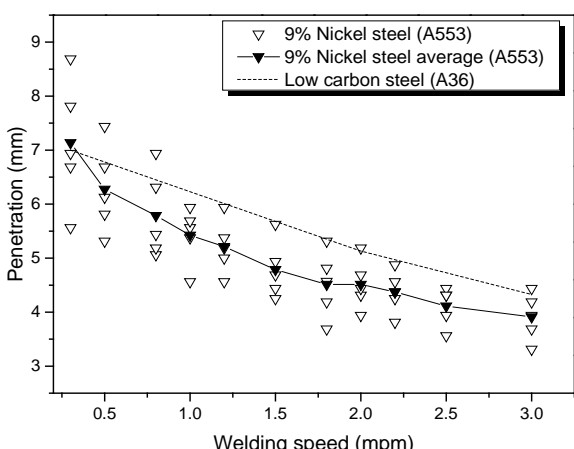

**Figure 15.** Penetration fluctuation of surface beads, according to welding speed.

The under-fill is reported to be caused by the synergy of the physical phenomenon of the molten part [24]. This is due to the complexities of the melt, such as volumetric shrinkage, surface tension, gravity, vapor pressure and phase transformation [25]. In this study, the cause of under-fill is explained by the phenomenon of the flow path of the exposure plasma vapor generated in the molten part. Although the steam generated by vaporization or organic plasma must be discharged to the outside

through the occurrence of the keyhole of the welded portion, the discharge channel is clogged by the fast welding speed. Consequently, the molten pool explosion (spatter) is generated, and the surface depression cannot be filled again. To prevent weld under-fill, welding must ultimately be performed at a speed equal to or less than the constant speed. Figure 16 illustrates the under-fill phenomenon occurring in the high-speed section, when the welding power is constant at 3.5 kW and only the speed changes. Figure 17 is a graph that identifies the boundary of the under-fill zone, depending on the welding speed. It can be observed from this graph that the under-fill phenomenon occurs at a constant speed in all laser-power sections.

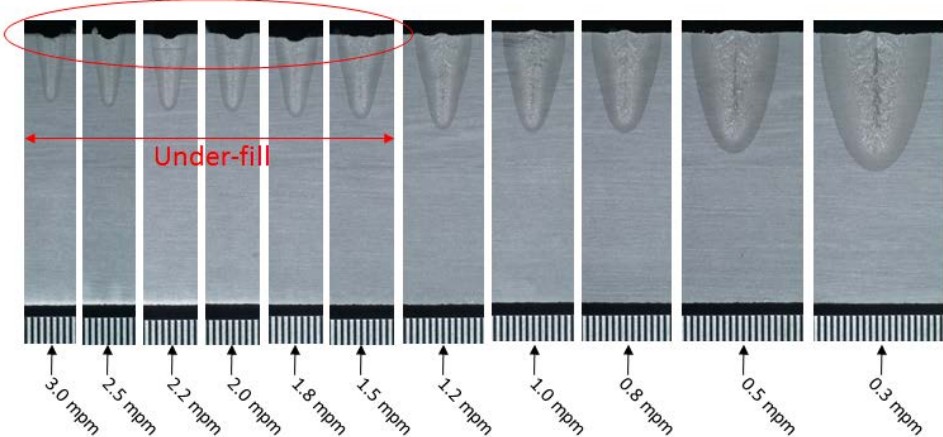

**Figure 16.** Under-fill phenomenon of A553-1 weld in high-speed section.

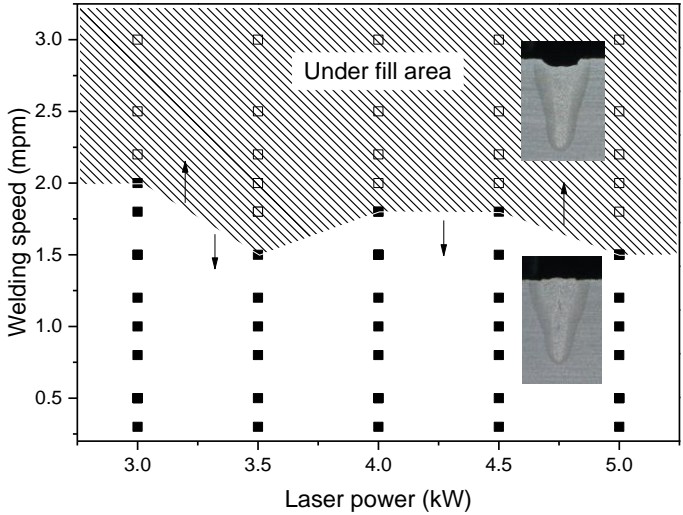

**Figure 17.** Under-fill section, according to welding speed.

## 4. Conclusions

In this study, the penetration shape of 9% nickel steel was confirmed by welding power and welding speed, which are the main laser-welding factors. The results of this study can be summarized as follows:

(1)　The increase in bead height and bead width of 9% nickel steel was not uniform as the laser power increased. However, the penetration depth of the two materials was highly dependent on the laser power, despite the difference in properties of the two materials. In the case of A553-1, when the welding power increased from 3 to 5 kW, the average penetration depth increased from 4.3 to 5.7 mm, according to Figure 11.

(2)  In the case of laser welding of A553-1, as the welding speed increased, the bead height, width and penetration depth were reduced. To prevent weld under-fill, welding must ultimately be performed at a speed equal to or less than the constant speed. It was confirmed that the occurrence of under-fill was dependent on the welding speed, and the result are observed in a specific range (over 1.5 m/min), according to Figure 17.

(3)  Based on the contents of laser-welding penetration of the A553-1 material obtained in this study, optimized conditions of laser welding will be studied. Moreover, we will verify weld zone by the mechanical test of yield, tensile and impact, under cryogenic conditions, in PART II of this study.

**Author Contributions:** "Conceptualization, J.K. (Jaewoong Kim) and J.K. (Jisun Kim); methodology, J.K. (Jaewoong Kim); software, J.K. (Jisun Kim); validation, J.K. (Jaewoong Kim); formal analysis, S.K.; resources, K.C.; data curation, S.K.; writing—original draft preparation, J.K. (Jaewoong Kim); writing—review and editing, J.K. (Jaewoong Kim); visualization, S.K. and K.C.; project administration, J.K. (Jaewoong Kim); All authors have read and agreed to the published version of the manuscript.

**Funding:** This research has been conducted with the support of the Ministry of Trade, Industry and Energy, Republic of Korea as "Development of small and medium LNG fuel storage module for coastal ship" in the material parts technology development project.

**Conflicts of Interest:** The authors declare no conflict of interest.

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
