# Peer review of "Laser Welding of ASTM A553-1 (9% Nickel Steel) (PART I: Penetration Shape by Bead on Plate)"

_metals, doi:10.3390/met10040484_

Round 1
Reviewer 1 Report
An interesting topic has been touched upon, but there are many questions about the text of the manuscript.
It would be more relevant and illustrative to compare the behavior of A553 and A36 steels in the BOP test using the same welding regimes (welding speed and laser power). And so it is not clear what is scientific novelty of paper, described results, and are expected by everybody.
Line 15, missed word “steel”, 9% nickel. In general, it is better to use steel A553, it looks bad when the sentence begins with a number, as on lines 47, 50.
Line 23, (101, 174), is not normally used in scientific articles. It is a third-person, impersonal narrative.
Line 28, which in keywords does “Flux cored arc welding”?
Lines 96, 103, 24, 65, the decoding of the abbreviation should be applied once, (means BOP), the HAZ too.
Line 112, which means.... 15t …..
Line 148, which means …. optimal microscope ….. It is clear that is meant …optical…, please specify which mark and model of microscope.
As for one of the main objects under study – the shape of the remelted zone and its geometric dimensions. I know from personal experience that with a decrease of a welding speed (0.5 mpm), the cross section shape will not be stable over the entire length of the remelted zone. With increasing speed, cross sectional stability will increase, approximately at speeds of 1.5 - 3 mpm. Also, as a rule, when not fully melted at different speeds, the liquid metal does not behave stably, pores are formed, which also affect the shrinkage or vice versa the height of the bead. For this research could be given more attention, which I did not see in this manuscript.
Two sentences on lines 198 – 203 are not clear to me.
Figure 11, if there's defocus: 0 everywhere, why specify it?
Instead of under-fill, it's better to use undercut.
On lines 266 – 267, the authors are absolutely correct in stating the reasons for undercutting and shrinking.
The conclusions are not clear, at least the experimental results could have been given in numbers.
Author Response
Thank you for your high opinion. Please see the attachments.

Reviewer 2 Report
The paper deals with the relevant topic regarding the application of highly effective welding processes such as laser welding in the LNG sector of the industry. This is a well structured manuscript. However, there are some suggestions for improving the scientific relevance of the paper:
- the applicability of laser welding technology for joining of cryogenic 9% Ni steel (X8Ni9, SA553) was recently examined by various research groups. As example, results were published by the authors A.M. El-Bathgy (CMRDI Agypt), M. Rethmeier (BAM Berlin), Yue Wu (Shanghai Jiao Tong University, Shanghai, China). The work presented does not refer to any existing studies on laser beam welding of 9% Ni steel. It is therefore recommended to complete the state of the art.
- line 19: The first is to use a welding electrode that is 20 times higher than the base metal....
What does 20 times higher mean? Please rewrite the sentence.
- line 112: The dimension of the each BOP test is 150mm x 150 mm x 15t. what does 15t mean?
- Table 2. The chemical composition of 9% nickel steel. You show a range of Ni content 7.5 - 8.5%. Does it range meet the requirements of the standard or have you carried out your own analysis? Please name the standard if applicable.
- line 69-70: The focus position is 148.8 mm, and it has a depth of focus of about 6 mm. Do you mean the focal length and the Rayleigh length? If relevant, please bring the specific terms in accordance with the conventional terms.
- cross sections in Fig. 15: In the case of a low welding speed and an increasing penetration depth, a centerline crack can be seen. These effects are known for partial penetrated laser beam welding, but these effects are not mentioned / discussed in the paper presented.
- A number of measurements were carried out in the frame of the work. Can you indicate the measurement errors and the statistical reliability of the measurements? The issue is, that the measuring on the individual cross section represents just current situation of the welding process and the penetration depth can fluctuate along the weld length. Is it possible to show some representative longitudinal sections?
With regard to the weld bead width and weld bead height, would it be possible to use an optical Profilometer?
Author Response

(The authors gave the same response as above.)

Round 2
Reviewer 1 Report
Reviewer comments (4) : which in keywords does “Flux cored arc welding”?
Answers and corrections: In response to your comments, it is revised.
Reviewer comments v2: Apparently, the authors misunderstood me. I meant that the manuscript is about laser welding, and I do not understand why the authors indicate "Flux Cored Arc Welding" in the keywords.
Figure 7. Crack, porosity, specter and under-fill of A553-1 after laser welding
Reviewer comments v2: Figure 7 (left side) crack instead of creack.
Reviewer comments (7) : Two sentences on lines 198 – 203 are not clear to me.
Answers and corrections: In response to your comments, the contents are added. Line 220-225
- Thermal conductivity coefficients of A553-1 and A36 are 17.43 J/m․sec°C and 52.7 J/m․sec°C, respectively. When heat source is supplied to material of A36 having a relatively high thermal conductivity characteristic, heat is diffused in all directions of the base material. It accelerates the continuous penetration of the heat source in the thickness direction of the base material. However, 9% nickel steel (A553) has a low thermal diffusion rate, and as the energy source is concentrated on the surface, the width of the surface beads increases.
Reviewer comments v2: In my opinion, in the context …. heat is diffused ….., …a low thermal diffusion rate…. the term diffusion doesn't fit. I leave it to the authors.
Reviewer comments (9) : Instead of under-fill, it's better to use undercut.
Answers and corrections: The AWS B1.11 Guide for Visual Inspection, provided by AWS, states that the under-fill is a defect in the shape of the weld root or face surface that is less than the thickness of the base material. It is commonly used in welding with Groove, but it can also be used in BOP test. Under cut occurs at the boundary between the base material and the melting zone. In this study, the under fill was judged to be an appropriate term because it was caused by the lack of the melt due to scattering in the center of the melt.
Reviewer comments v2: Answer accepted, I apologize for not fully understanding the terminology.
Reviewer comments (10) :On lines 266 – 267, the authors are absolutely correct in stating the reasons for undercutting and shrinking.
Answers and corrections: It is deleted.
Reviewer comments v2: Why authors have removed these sentences, I mean, that's good knowledge of the authors. Or is it the wish of another reviewer?
Reviewer comments (11) : The conclusions are not clear, at least the experimental results could have been given in numbers.
Answers and corrections: In response to your comments, the conclusions are revised. (Line 308-317)
- -> (1) The increase in bead height and bead width of 9% nickel steel is not uniform as the laser power increased. But the penetration depth of the two materials is highly dependent on the laser power, despite the difference in properties of the two materials. In the case of A553-1, when the welding power increases from 3kW to 5Kw, the average penetration depth increases from 4.3 mm. to 5.7 mm.
- (2) In case of laser welding of A553-1, as the welding speed increases, the bead height, width and penetration depth are reduced. To prevent weld under-fill, welding must ultimately be performed at a speed equal to or less than the constant speed. It was confirmed that the occurrence of under-fill was dependent on the welding speed and the result are observed in a specific range (over 1.5mpm).
Reviewer comments v2: ….from 3kW to 5Kw…… misprint.

Author Response
"Please see the attachment."

Reviewer 2 Report
Please consider the following comments and suggestions regarding the revised version of the manuscript:
Suggestion for Line 56: Table 2. The chemical composition of ASTM A553 Type 1 9% nickel steel
Line 112: N2
Line 163: HNO3
Line 114: The dimension of the each BOP test is 150mm x 150 mm x 15mm (thickness), but in the Line 150 we can see: In this study, the size of welding specimen is 150 mm x 300 mm x 15mm. Which sample size is correct?
Line 164: What does an optimal microscope mean? Do you mean optical microscope?
Line 170: My opinion as a reader is that showing a simply picture of the microscope without naming its characteristics does not necessarily fit the scientific paper format, because it is more or less a standard measuring device. Please provide some technical data on the optical measuring system: type, magnification x ...
Line 173: Measured equipment and method for bead geometry ?
Suggestion: Equipment and method for measuring....
Line 199: scatter or spatter? please check
Line 200: Creack?
Line 201: specter?
Author Response
"Please see the attachment."
